# LSTM-Based Projectile Trajectory Estimation in a GNSS-Denied Environment [note 1]

**DOI:** 10.3390/s23063025

**Published:** 2023-03-10

**Authors:** Alicia Roux, Sébastien Changey, Jonathan Weber, Jean-Philippe Lauffenburger

**Affiliations:** 1French-German Research Institute of Saint-Louis, 5 Rue du Général Casssagnou, 68300 Saint-Louis, France; sebastien.changey@isl.eu; 2Institut de Recherche en Informatique, Mathématiques, Automatique et Signal (IRIMAS), Université de Haute-Alsace, 2 Rue des Frères Lumière, 68100 Mulhouse, France; jonathan.weber@uha.fr (J.W.); jean-philippe.lauffenburger@uha.fr (J.-P.L.)

**Keywords:** long-short-term-memory, projectile trajectory, navigation

## Abstract

This paper presents a deep learning approach to estimate a projectile trajectory in a GNSS-denied environment. For this purpose, Long-Short-Term-Memories (LSTMs) are trained on projectile fire simulations. The network inputs are the embedded Inertial Measurement Unit (IMU) data, the magnetic field reference, flight parameters specific to the projectile and a time vector. This paper focuses on the influence of LSTM input data pre-processing, i.e., normalization and navigation frame rotation, leading to rescale 3D projectile data over similar variation ranges. In addition, the effect of the sensor error model on the estimation accuracy is analyzed. LSTM estimates are compared to a classical Dead-Reckoning algorithm, and the estimation accuracy is evaluated via multiple error criteria and the position errors at the impact point. Results, presented for a finned projectile, clearly show the Artificial Intelligence (AI) contribution, especially for the projectile position and velocity estimations. Indeed, the LSTM estimation errors are reduced compared to a classical navigation algorithm as well as to GNSS-guided finned projectiles.

## 1. Introduction

Projectile navigation is mainly based on IMU (Inertial Measurement Unit) and GNSS (Global Navigation Satellite System) measurements due to the high dynamic constraints imposed on projectiles and the low-cost sensor requirements. Classically, IMU and GNSS measurements are combined with Kalman Filters to estimate a trajectory. The IMU measurements are integrated to predict the trajectory in order to be corrected by the GNSS receiver measurements [1,2,3,4]. Nevertheless, GNSS signals are not always available due to the environment configuration and are vulnerable to jamming and spoofing [5,6,7]. For this purpose, users aim to exclude these measurements for trajectory estimation [8,9,10].

Moreover, Artificial Intelligence (AI) is increasingly used for defense applications such as surveillance, reconnaissance, tracking or navigation [11,12,13,14,15]. Indeed, AI is an interesting approach to correct model approximations, to limit the influence of incomplete or incorrect measurements or to determine complex models from system data. Therefore, this paper presents an AI-based projectile trajectory estimation method in a GNSS-denied environment using only the embedded IMU and pre-flight parameters specific to the ammunition.

Considering that a trajectory is a time series, AI provides interesting approaches for its estimation. Indeed, Recurrent Neural Networks (RNNs) are perfectly adapted to time series prediction as illustrated in [16,17,18]. RNNs are composed by feedback loops, i.e., they memorize past data through hidden states to predict future data. However, the simplest form of RNNs exhibits convergence issues during the training step such as vanishing or exploding gradient problems. So another form of recurrent network is considered: the Long Short-Term Memory (LSTM) [19,20]. A LSTM includes a memory cell in addition to hidden states, in order to capture both long-term and short-term time dependencies.

This paper presents an AI-based solution to estimate a projectile trajectory in a GNSS-denied environment. In summary, the main contributions of this work are:•to detail an LSTM-based approach to estimate projectile positions, velocities and Euler angles from the embedded IMU, the magnetic field reference, pre-flight parameters and a time vector.•to present BALCO (BALlistic COde) [21] used to generate the dataset. This simulator provides true-to-life trajectories of several projectiles according to the ammunition parameters.•to investigate different normalization forms of the LSTM input data in order to evaluate their contribution on the estimation accuracy. For this purpose, several LSTMs are trained with different input data normalizations.•to study the impact of the local navigation frame rotation on the estimation accuracy. Rotating the local navigation frame during the training step allows having similar variation ranges along the three axes, especially for the lateral position, which is extremely small compared to the two other axes. This method shares the same goals as normalization but without any information loss.•to examine the influence of inertial sensor models on estimation accuracy. For this purpose, two inertial sensor error models are studied in order to evaluate their influence on LSTM predictions.•to compare the LSTM accuracy to a Dead-Reckoning, performed on finned mortar trajectory. Estimation methods are evaluated through error criteria based on the Root Mean Square Error and the impact point error.

The outline of the paper is as follows. Section 2 presents an introduction to projectile navigation, AI applications in the military field and to the LSTM basics. Section 3 focuses on the projectile trajectory dataset and LSTM specifications. Section 4 presents the data pre-processing; the input data normalization and the local navigation frame rotation during training. Finally, Section 5 presents projectile trajectory estimation results by analyzing the influence of the local navigation frame rotation, input data normalization and sensor model on estimation accuracy.

## 2. Related Work

This part presents the traditional projectile navigation methods and the sensors used, the applications of artificial intelligence for defense and the LSTM operating principle.

### 2.1. Model-Based Projectile Trajectory Estimation

Projectile navigation requires sensors able to resist to extreme conditions (acceleration shocks around 50,000 g along the longitudinal axis, high rotation rates around 15,000 rpm for a 155 mm shell) as well as to be relatively small and inexpensive due to the space and cost constraints imposed on projectiles [22,23,24]. For this purpose, projectile navigation mainly uses IMUs (Inertial Measurement Units) composed by accelerometers, gyrometers, and magnetometers as well as GNSS (Global navigation satellite system) data. On one hand, the IMU measurement integration, performed at high frequency (∼1000 Hz), provides an accurate short-term projectile trajectory estimation, but deviates at long term due to sensor drift [1]. On the other hand, GNSS receivers provide accurate long-term position information at a significantly lower frequency (∼10 Hz), but can be easily spoofed and jammed [5,6,7]. Due to their evident complementarity, IMUs and GNSS are classically fused by different types of Kalman filters for trajectory estimation, as in [2,3,4].

One challenge with high-speed spinning projectiles is to accurately estimate altitudes. Depending on projectile specifications, different methods can be considered, as in [25,26] by exploiting GNSS signals or in [9,22] by using accelerometers, gyrometers or magnetometers. Nevertheless gyrometers are commonly omitted because, under cost limits, many of them saturate and do not resist to launch phases. Therefore, magnetometers, less expensive and able to resist to high accelerations, are often used [27,28,29] with different kinds of Kalman filters (Extended Kalman Filter, Adaptive Extended Kalman Filter, Mixed Extended-Unscented Filter).

As mentioned in the introduction (see Section 1), GNSS signals are easily spoofed and jammed. Therefore, more and more, approaches are proposed in a GNSS-denied environment in order to estimate a projectile trajectory, using only inertial measurements, as in [8,30,31,32].

### 2.2. AI-Based Trajectory Estimation

AI methods are increasingly used in the military field especially for:*Surveillance and target recognition* where machine learning algorithms applied to computer vision detect, identify and track objects of interest.For example, the Maven project, presented by the US Department of Defense (DoD), focuses on automatic target identification and localization from images collected by Aerial Vehicles [11].*Predictive maintenance* to establish the optimal time to change a part of a system, as the US Army does on F-16 aircraft [12].*Military training* where AI is used in training simulation software to improve efficiency through various scenarios, such as AIMS (Artificial Intelligence for Military Simulation) [13].*Analysis and decision support* to extract and deal with relevant elements in an information flow, to analyze a field or to predict events. The Defense Advanced Research Projects Agency (DARPA) aims to equip US Army helicopter pilots with augmented reality helmets to support them in operations [14].*Cybersecurity*, as military systems are strongly sensitive to cyberattacks leading to loss and theft of critical information. To this end, the DeepArmor program from SparkCognition uses AI to protect, detect and block networks, computer programs and data from cyber threats [15].

Although widely integrated in the military development programs, AI is relatively uncommonly applied to projectile trajectory estimation although some kinds of networks such as recurrent networks are perfectly adapted to this task.

#### 2.2.1. Recurrent Neural Networks (RNNs)

Recurrent Neural Networks (RNNs) are a class of neural networks perfectly adapted for time series prediction [19,20] as they exploit feedback loops, i.e., an RNN cell output at the previous time is used as input at the current time.

As illustrated in Figure 1, an RNN exploits a time series $x=\left[x_{0},x_{1},\ldots ,x_{\tau }\right]\in R^{\tau \times F}$ of length τ with *F* input features to predict an output *y*. Each prediction $y_{t}$ is determined by one RNN cell from the current input $x_{t}$ and the previous output $h_{t-1}$, also called hidden state, which memorizes the past information [19,20].

Vanilla RNN, the simplest RNN structure, suffers from gradient vanishing and explosion during the training step [33,34]. In the gradient vanishing case, backpropagation from the last layer to the first layer leads to a gradient reduction. Then, the first layer weights are no longer updated during training and the Vanilla RNN does not learn any features. In the gradient explosion case, gradients become increasingly large leading to huge weight updates and thus resulting in Vanilla RNN divergence.

Moreover, to predict $y_{t}$ at timestamp *t*, the Vanilla RNN uses only the input $x_{t}$ at the current time and the hidden state $h_{t-1}$ at the previous time, containing short-term past features. For this reason, Vanilla RNN is ineffective to memorize long-term past events. To overcome these issues, memory cells are added to the Vanilla RNN, forming the Long Short-Term Memory (LSTM) [19].

#### 2.2.2. Long Short-Term Memory Cell

Based on the recurrent network overview presented above, this paragraph focuses only on the LSTM cell operating principle. An LSTM is composed by several cells to deal with short and long-term memory. As shown in Figure 2, to predict $y_{t}$ at timestamp *t*, an LSTM uses the input data $x_{t}$ at the current time, the hidden state $h_{t-1}$ at the previous time to memorize short-term past events, and the memory cell state $c_{t-1}$ at the previous time to memorize the long-term past events. An LSTM cell is composed by three gates:–the *forget gate* filters, through a Sigmoid function σ, data contained in the concatenation of $x_{t}$ and $h_{t-1}$. Data are forgotten for values close to 0 and are memorized for values close to 1. The *forget gate* model is:
(1)$$\begin{array}{c} f_{t}=\sigma \left(W_{f}.\left[h_{t-1},x_{t}\right]+b_{f}\right) \end{array}$$–the *input gate* extracts relevant information from $\left[h_{t-1},x_{t}\right]$ by applying a Sigmoid σ and a Tanh function. The *input gate* is represented by the following:
(2)$$\begin{array}{c} \begin{array}{cccc} i_{t}=\sigma \left(W_{i}.\left[h_{t-1},x_{t}\right]+b_{f}\right) & & & {\tilde{C}}_{t}=tanh\left(W_{c}.\left[h_{t-1},x_{t}\right]+b_{c}\right). \end{array} \end{array}$$The memory cell $c_{t}$ is updated from the *forget gate*$f_{t}$ and the *input gate* $i_{t}$ and ${\tilde{C}}_{t}$, to memorize pertinent data:
(3)$$\begin{array}{c} c_{t}=f_{t}\times c_{t-1}+i_{t}\times {\tilde{C}}_{t} \end{array}$$–the *output gate* defines the next hidden state $h_{t}$ containing information about previous inputs. The hidden state $h_{t}$ is updated with the memory cell $c_{t}$ normalized by a Tanh function and $\left[h_{t-1},x_{t}\right]$ normalized by a Sigmoid function:
(4)$$\begin{array}{c} h_{t}=\sigma \left(W_{h}.\left[h_{t-1},x_{t}\right]+b_{h}\right)\times tanh\left(c_{t}\right) \end{array}$$
with $W_{\left(.\right)}$ and $b_{\left(.\right)}$, the different gate weights and biases.

Currently, few works have appied recurrent networks in the military context. They are commonly used for aircraft navigation [16,17], vehicle trajectory estimation [35], maritime route prediction [36] or human motion prediction [18,37]. It is, however, interesting to mention [38], which focused on projectile trajectory estimation based on LSTMs trained from incomplete and noisy radar measurements.

## 3. Problem Formulation

This part presents the projectile fire simulation dataset generated by BALCO (BALlistic COde) [21] and the LSTM input data used to estimate projectile trajectories.

### 3.1. The Projectile Trajectory Dataset BALCO (BALlistic COde)

Results reported in this paper exploit a projectile fire dataset generated by BALCO [21,39], i.e., a high fidelity projectile trajectory simulator based on motion equations with six to seven degrees of freedom and discretized by a seventh order Runge-Kutta method. BALCO enables the consideration of different earth models (flat earth, spherical, ellipsoidal), different atmospheric models (standard atmosphere or defined by the user) or different aerodynamic models (axisymmetric or non-axisymmetric projectiles, aerodynamic coefficients described in correspondence tables or polynomials). BALCO accuracy is validated in comparison to the reference program PRODAS (Projectile Rocket Ordnance Design and Analysis System) by considering different projectile types, various initial conditions and different meteorological conditions.

In order to estimate projectile trajectories, three reference frames are considered.

The *local navigation frame **n*** (black frame in Figure 3) tangent to the Earth and assumed fixed during the projectile flight.

The *body frame **b*** (red frame in Figure 3), which is an ideal hypothetical frame placed exactly at the projectile gravity center, in which the IMU must be placed, providing perfect inertial measurements.

The *sensor frame **s*** (green frame in Figure 3) rigidly fixed to the projectile and misaligned with the projectile gravity center, considered as the frame where the inertial measurements are performed.

Results reported in this paper are applied to the estimation of a finned mortar trajectory. The finned projectile dataset, generated by BALCO, includes 5000 fire simulations and where each one includes:the **inertial measurements in the body frame**
***b***** and in the sensor frame**
***s***, i.e., gyrometer $\omega \in R^{3}$, accelerometer $a\in R^{3}$ and magnetometer $h\in R^{3}$ measurements. Three kinds of inertial measurements are available:–the *Perfect IMU measurements* performed in the body frame ***b*** (red frame in Figure 3), in the ideal case where all the three inertial sensors are perfectly aligned with the projectile gravity center and where no sensor default model is taken into account providing ideal inertial measurements, i.e., without any noise or bias. These measurements are not exploited in this work but are necessary to provide realistic inertial data.–the *IMU measurements* performed in the sensor frame ***s*** (green frame in Figure 3): issued from the *Perfect IMU measurements* where a sensor error model is added. This error model, specific to each sensor axis, includes a misalignment, a sensitivity factor, a bias and a noise (assumed zero mean white Gaussian noise). Thus, this measurement accurately models an IMU embedded in a finned projectile.–the *IMU DYN measurements* performed in the sensor frame ***s*** (green frame in Figure 3): issued from *IMU measurements* to which a transfer function is added to each sensor. For each sensor, *IMU DYN measurements* are modeled by:
(5)$$\begin{array}{c} y_{sensor{,}_{IMU\;DYN}}=\frac{1}{1+as+bs^{2}}y_{sensor{,}_{IMU}} \end{array}$$
with $y_{sensor{,}_{IMU}}$ the *IMU measurements* of the considered sensor, $y_{sensor{,}_{IMU\;DYN}}$ the corresponding *IMU DYN measurements* and with *a* and *b*, the coefficients of the sensor transfer function defined via BALCO. This sensor model allows to model the response of the three sensors over the operating range.the **magnetic field reference $h_{n}\in R^{3}$** in the local navigation frame ***n***, assumed constant during the projectile flight.**flight parameters**, which are, in the case of a finned projectile, the fin angle $\delta_{f}$ to stabilize projectiles, the initial velocity $v_{0}$ at barrel exit and the barrel elevation angle α, relatively important to obtain ballistic trajectories with short ranges.a **time vector $k\Delta_{t}$** where $\Delta_{t}$ is the IMU sampling period: $\Delta_{t}=1\times 10^{-3}s$.the **reference trajectory**, i.e., the projectile position $p\in R^{3}$, velocity $v\in R^{3}$ and Euler angles $\Psi \in R^{3}$ in the local navigation frame ***n*** at the IMU frequency. This trajectory is used to evaluate the LSTMs accuracy and to compute errors.

### 3.2. Data Characteristics and LSTM Requirements

The LSTM predictions at time *t* are obtained from three-dimensional input data of size $\left(B_{atch\;size},S_{eq\;len},I_{n\;Features}\right)$, with $B_{atch\;size}$ the number of sequences considered, $S_{eq\;len}$ the number of time steps in the sequence and $I_{n\;Features}$ the number of features describing each time step. The input features are $I_{n\;Features}=\left(M,P,T\right)\in R^{16}$, such as the following:-$M\in R^{12}$ *the inertial data*, including *IMU* or *IMU DYN measurements* in the sensor frame ***s*** and the reference magnetic field $h_{n}\in R^{3}$ in the local navigation frame ***n*** presented in Section 3.1,-$P\in R^{3}$ *the flight parameters*. In the case of a finned projectile, the three flight parameters are the fin angle $\delta_{f}$, the initial velocity $v_{0}$ and the the barrel elevation angle α.-$T\in R^{1}$ *the time vector*, such as $T=k\Delta_{t}$ with *k* the considered time step and $\Delta_{t}$ the IMU sampling period.

Various LSTMs are trained and differ depending on the output features learned. Indeed, according to the input data of size $\left(B_{atch\;size},S_{eq\;len},I_{n\;Features}\right)$, LSTMs estimate a projectile trajectory modeled by a vector of size $\left(B_{atch\;size},O_{ut\;Features}\right)$ and where $O_{ut\;Features}$ represents the number of output features. The output features $O_{ut\;Features}$ are 9 or 3, depending on the type of LSTM considered. Thus, the following notations are used:-$LSTM_{ALL}$ trained to estimate 9 output features which are the projectile position $p\in R^{3}$, velocity $v\in R^{3}$ and Euler angles $\Psi \in R^{3}$ in the navigation frame ***n***.-$LSTM_{POS}$ trained to estimate 3 output features, which are the projectile position $p\in R^{3}$ expressed in the navigation frame ***n***.-$LSTM_{VEL}$ trained to estimate 3 output features which are the projectile velocity $v\in R^{3}$ expressed in the navigation frame ***n***.-$LSTM_{ANG}$ trained to estimate 3 output features which are the projectile Euler angles $\Psi \in R^{3}$ in the navigation frame ***n***.

## 4. LSTM Input Data Preprocessing

This section details the two input data pre-processing methods in order to study their influence on estimation accuracy. To manage the different projectile dynamics along the three navigation axes, two data preprocessing methods are investigated: the LSTM input data normalization and local navigation frame rotation allowing to rescale each component of a 3D value on similar variation ranges. To this end, LSTMs presented in Section 3.2 are declined in 8 versions, reported in Table 1, to study the influence of the Min/Max MM(.) and the Standard Deviation STD(.) normalization, and the local navigation frame rotation on estimation accuracy.

### 4.1. LSTM Input Data Normalization

Network input data normalization is a preprocessing data approach to rescale input data on similar variation ranges while preserving the same distribution and ratios as the original data. Input data normalization is used to prevent some input data features from having a greater influence than other features during training and to improve gradient backpropagation convergence. In other words, input data with different ranges can lead to lower network estimation performance. The small input values have a small influence during prediction, and therefore the network weights are updated according to the high input values, which can lead to a significant network weight update and therefore a slower network convergence or the network convergence to a local minimum.

According to Table 1, two kinds of normalization are used:*Min/Max normalization MM(.): *$x_{MM}=\frac{x-x_{min}}{x_{max}-x_{min}}$ with $x_{max}$ and $x_{min}$ the maximum and minimum of *x*.*Standard Deviation normalization STD(.): *$x_{STD}=\frac{x-\mu_{x}}{\sigma_{x}}$ with *x* the quantity to normalize, $\mu_{x}=\mu \left(x\right)$ its mean and $\sigma_{x}=\sigma \left(x\right)$ its standard deviation. Thus, $x_{STD}$ is a quantity with a zero-mean and a standard deviation of one.

In order to study the impact of input data normalization on estimation accuracy, versions $V_{2}$ and $V_{4}$ use normalization by features while versions $V_{3}$, $V_{5}$, $V_{7}$ and $V_{8}$ use normalization for all features. Moreover, the normalization factors $x_{max}$, $x_{min}$, $\mu_{x}$ and $\sigma_{x}$ are computed before the training step on the training dataset, as in the following:(6)$$\begin{array}{ccc} & x_{max}=\frac{1}{N_{sim}}\sum_{i=1}^{N_{sim}}max\left\{\chi_{i}\right\},\;\;\;\;x_{min}=\frac{1}{N_{sim}}\sum_{i=1}^{N_{sim}}min\left\{\chi_{i}\right\} & \end{array}$$(7)$$\begin{array}{cc} & \mu_{x}=\frac{1}{N_{sim}}\sum_{i=1}^{N_{sim}}\mu \left(\chi_{i}\right),\;\;\;\;\sigma_{x}=\frac{1}{N_{sim}}\sum_{i=1}^{N_{sim}}\mu \left(\chi_{i}\right) \end{array}$$
with $N_{sim}$ the number of simulation in the training dataset and with $\chi_{i}$ the considered quantities of the simulation n° i, which are $\chi_{i}=\left[\begin{array}{ccc} M & P & T \end{array}\right]$ for versions $V_{3}$, $V_{5}$, $V_{7}$ and $V_{8}$, and $\chi_{i}=M$ or $\chi_{i}=P$ or $\chi_{i}=T$ for versions $V_{2}$ and $V_{4}$.

### 4.2. Local Navigation Frame Rotation

The local navigation frame rotation aims to rotate the local navigation frame ***n*** by a fixed angle γ (local rotated navigation frame $n_{\gamma }$), such as $x_{\gamma }=R_{\gamma }x$ with $x\in R^{3}$ defined in ***n***, $x_{\gamma }\in R^{3}$ expressed in $n_{\gamma }$ and $R_{\gamma }\in SO\left(3\right)$ the transition matrix from the local navigation frame ***n*** to the local rotated navigation frame $n_{\gamma }$ as in the following:(8)$$R_{\gamma }=[\begin{array}{ccc} cos(\gamma ) & -sin(\gamma ) & 0\\ sin(\gamma ) & cos(\gamma ) & 0\\ 0 & 0 & 1 \end{array}]\;[\begin{array}{ccc} cos(\gamma ) & 0 & sin(\gamma )\\ 0 & 1 & 0\\ -sin(\gamma ) & 0 & cos(\gamma ) \end{array}]\;[\begin{array}{ccc} 1 & 0 & 0\\ 0 & cos(\gamma ) & -sin(\gamma )\\ 0 & sin(\gamma ) & cos(\gamma ) \end{array}].$$

The navigation frame rotation allows a quantity $x\in R^{3}$ expressed in the navigation frame ***n*** to modify its three components in order to have a similar amplitude order for the three components. The local navigation frame rotation provides similar variation ranges of a quantity along the three axes. This approach is used to ensure that the LSTMs adequately estimate a quantity with small magnitudes along one axis, even though the variations are considerably larger along the other two axes.

As illustrated in Figure 4, the variation range of the projectile position along the y-axis is significantly smaller than along the x and z-axes and thus, the expressed position in the local rotated navigation frame $n_{\gamma }$ provides similar amplitudes along the three axes. For example, projectile position variation ranges along the x and z-axes are around several kilometers, while the position along the y-axis varies by a few meters. As illustrated in Figure 4, expressed position in the rotated navigation frame $n_{\gamma }$ provides similar amplitudes along the three axes.

All quantities expressed in the local navigation frame are rotated, i.e., the projectile position *p*, velocity *v* and Euler angles Ψ. Moreover, the angle γ is fixed for all trajectories in the dataset and is determined according to the data used in this paper and is the same to express the position, velocity or Euler angles in the rotated navigation frame. This angle is determined according to the data used in this paper in particular to have similar magnitudes for the three positions, as for the velocity. During the training step, labels are expressed in the local rotated navigation frame $n_{\gamma }$ and LSTMs predict trajectories in $n_{\gamma }$. During the test step, LSTMs estimate projectile trajectories in the local rotated navigation frame $n_{\gamma }$, and then, estimations are moved back to the initial local navigation frame ***n***.

## 5. Results and Analysis

This part of the paper reports LSTM results applied to finned projectiles. A first section focuses on the influence of the normalization and the local navigation frame rotation on the estimation accuracy for short training. A second section validates LSTMs on a large dataset by focusing on the impact of the local navigation frame rotation and the IMU model.

The LSTMs’ performances are evaluated in comparison to a classical navigation algorithm, i.e., a Dead-Reckoning. This algorithm integrates gyrometer ω and accelerometer *a* measurements to estimate at each discrete time *k*:(9)$$\begin{array}{c} \begin{array}{c} R_{k}=R_{k-1}{\left[\omega_{k}\Delta_{t}\right]}_{\times },\\ v_{k}=v_{k-1}+\left(R_{k-1}a_{k}+g\right)\Delta_{t},\\ p_{k}=p_{k-1}+v_{k-1}\Delta_{t}+\frac{1}{2}\left(R_{k-1}a_{k}+g\right)\Delta_{t}^{2}, \end{array} \end{array}$$
with $R_{k}\in SO\left(3\right)$ the rotation matrix from the sensor frame ***s*** to the local navigation frame ***n***, $g\in R^{3}$ the constant gravity vector, $p_{k}\in R^{3}$ and $v_{k}\in R^{3}$, respectively, the projectile position and velocity, and ${\left[.\right]}_{\times }$ the skew matrix. This algorithm is generally used for Kalman filters for the prediction step to estimate trajectory, as presented in [2,3,4,8,24,40].

### 5.1. Impact of the Input Data Normalization and the Local Navigation Frame Rotation

This section reports the estimated trajectories of a finned projectile according to $LSTM_{ALL}$, $LSTM_{POS}$, $LSTM_{VEL}$ and $LSTM_{ANG}$, each declined in the 8 versions $V_{1}$–$V_{8}$ presented in Section 3.2. The training parameters are summarized in the Table 2.

$S_{eq\;len}$ corresponds to 20 samples representing a window of 0.02 s as the sensor sampling period is $\Delta_{t}=1\times 10^{-3}\;s$. This parameter is adjusted according to the input data used.

#### 5.1.1. Qualitative Validation: One Finned Projectile Fire Simulation

Figure 5 focuses on the estimated positions and orientation for one projectile shot. For readability reasons, three estimation methods are first compared: the Dead-Reckoning (Equation 9), $LSTM_{ALL,V_{1}}$ and $LSTM_{ALL,V_{6}}$ (local navigation frame rotation).

As shown in Figure 5, positions estimated by the LSTMs are significantly more accurate than the Dead-Reckoning. Nevertheless, LSTMs are only accurate in estimating the pitch and yaw angle. Errors in the roll angle estimation are due to the finned projectile rotation rate. The LSTMs fail to fully capture the roll angle dynamics. Moreover, the local navigation frame rotation improves projectile position estimation but slightly degrades pitch angle estimation.

#### 5.1.2. Quantitative Evaluation: Analysis on the Whole Test Dataset

To validate the previous observations, $LSTM_{ALL,\;POS,\;VEL,\;ANG,\;V_{1-8}}$ are evaluated on the $N_{sim}$ simulations in the test dataset according to two criteria based on the Root Mean Square Error (RMSE) defined as $RMSE_{x}=\sqrt{\frac{1}{N}\sum_{k=1}^{N}{\left(x_{k,ref}-{\hat{x}}_{k}\right)}^{2}}$, with $\hat{x}$ the estimate, $x_{ref}$ the reference and *N* the number of samples for one simulation.

The two evaluation criteria are as follows:*Success Rate $C_{1}$*: number of simulations where a LSTM RMSE is strictly smaller than the Dead-Reckoning.*Error Rate $C_{2}$*: percentage of LSTM error compared to Dead-Reckoning errors.
(10)$$\begin{array}{c} \begin{array}{ccc} C_{1}=\sum_{k=1}^{N_{sim}}RMSE_{LSTM}<RMSE_{DR}, & & C_{2}=\frac{100}{N_{sim}}\sum_{k=1}^{N_{sim}}\frac{RMSE_{LSTM}}{RMSE_{LSTM}+RMSE_{DR}} \end{array} \end{array}$$

The position, velocity and orientation success rate $C_{1}$ and the error rates $C_{2}$ are presented in Figure 6, Figure 7 and Figure 8. Figure 5, Figure 6, Figure 7 and Figure 8 has been modified for better readability.

*Position analysis results (see Figure 6):* The LSTMs outperform Dead-Reckoning for position estimation, especially along the y-axis. Normalizations affect position estimates differently as STD(T),STD(M),STD(P) ($V_{4}$) and MM(T,M,P) ($V_{3}$) are not appropriate to this application. In addition, the normalization leads to less accuracy as it implies a loss of information. Finally, rotating the local navigation frame improves the accuracy according to $C_{2}$ along the three axis.

*Veclocity analysis results (see Figure 7):* As previously, the LSTMs clearly outperform Dead-Reckoning for velocity estimation. Specialized networks $LSTM_{VEL}$ are a bit better than $LSTM_{ALL}$. The STD normalization for all features $V_{5}$ exhibits the best results among the different normalization options investigated, especially for velocity along the z-axis. Moreover, rotating the local navigation frame $V_{6}$ significantly improves the projectile velocity estimation along all the three axes.

*Euler angles analysis results (see Figure 8):* The LSTMs deteriorate the roll ϕ angle estimation compared to the Dead-Reckoning, but accurately estimate the yaw angle ψ. As previously, the STD(T,M,P) ($V_{5}$) normalization of $LSTM_{ALL}$ exhibits the best performances for the three Euler angles estimation as well as the local navigation frame rotation.

In summary, the LSTM end-to-end estimation is particularly appropriate for projectile position and velocity estimation. Moreover, from this evaluation study, it can be concluded that specialized networks do not significantly improve the estimation accuracy and STD(T,M,P) ($V_{5}$) normalization is more appropriate to estimate a projectile trajectory compared to other normalizations. Finally, rotating the local navigation frame is an efficient method to optimize projectile position and velocity estimation.

### 5.2. Impact of Inertial Measurement Type and Local Navigation Frame Rotation on Estimation Accuracy

The dataset presented in Section 3.1 contains two kinds of inertial readings; *IMU measurement*, used so far, and *IMU DYN measurement*, where sensors are characterized by a 2nd order model.

This section focuses on the impact of inertial data and navigation frame rotation on LSTM estimation accuracy. To this end, four LSTMs are trained to estimate positions, velocities and orientations of a finned projectile: $LSTM_{IMU,\;V_{1}}$ (no rotation), $LSTM_{IMU\;DYN,\;V_{1}}$ (no rotation), $LSTM_{IMU,\;V_{6}}$ (navigation frame rotation) and $LSTM_{IMU\;DYN,\;V_{6}}$(navigation frame rotation). LSTM specifications are given in Table 3.

#### 5.2.1. Impact of the Local Navigation Frame Rotation and IMU Measurement

This section focuses on $LSTM_{IMU,\;V_{1}}$ (no rotation), $LSTM_{IMU,\;V_{6}}$ (rotation) and the Dead-Reckoning algorithm (Equation 9) for position, velocity and Euler angles estimation. Network characteristics are presented in Table 3.

Figure 9 presents the average error distributions $\bar{e}$ and the corresponding standard deviations σ evaluated for positions and Euler angles, with the three estimation methods considered, such as
(11)$$\begin{array}{ccc} & \bar{e}=\frac{1}{N}\sum_{k=1}^{N}\left(x_{ref}-\hat{x}\right), & \end{array}$$
(12)$$\begin{array}{cc} & \sigma =\sqrt{\frac{1}{N}\sum_{k=1}^{N}{\left(\left[x_{ref}-\hat{x}\right]-\bar{e}\right)}^{2}}, \end{array}$$
where $x_{ref}$ is the reference, $\hat{x}$ the estimate and *N* the number of samples in the considered simulation.

According to Figure 9, the Dead-Reckoning mean error dispersion (red) is very large compared to those of LSTMs (green and blue). Thus, LSTMs are perfectly adapted to estimate finned projectile positions and velocities. Focusing on $p_{z}$, $v_{x}$ and $v_{z}$, the $LSTM_{IMU,\;V_{1}}$ average errors are not centered on zero compared to $LSTM_{IMU,\;V_{6}}$ (rotation). Thus, the local navigation frame rotation improves these estimates. As previously observed, the LSTMs fail to estimate the projectile roll angle even if the finned projectile rotation rate is low. Furthermore, the centering and dispersion of angle errors indicate that the LSTMs suffer to estimate the projectile orientation despite the yaw angle accuracy.

#### 5.2.2. Impact of the Local Navigation Frame Rotation and IMU DYN Measurement

Figure 10 presents the average error $\bar{e}$ (Equation 12) distributions for positions, velocities, and Euler angles estimated by $LSTM_{IMU\;DYN,\;V_{1}}$ (no rotation), $LSTM_{IMU\;DYN,\;V_{6}}$ (rotation), and the Dead-Reckoning (Equation 9).

According to Figure 10, the LSTMs accurately estimate the projectile position and velocity over a large dataset, despite dynamic inertial data causing, in contrast, the Dead-Reckoning divergence. Furthermore, the error distribution centering analysis allows to conclude that the local navigation frame rotation improves the estimation of $p_{y}$, $v_{x}$ and $v_{y}$. As for previous orientation estimation results, both LSTMs fail to estimate the projectile roll angle ϕ.

#### 5.2.3. Evaluation Metric

The performance of $LSTM_{IMU\;V_{1}}$, $LSTM_{IMU\;V_{6}}$, $LSTM_{IMU\;DYN\;V_{1}}$, $LSTM_{IMU\;DYN\;V_{6}}$, and the Dead-Reckoning (Equation 9) are evaluated using two evaluation criteria computed for each simulation in the test dataset:*Mean Absolute Error:*(13)$$\begin{array}{c} MAE=\frac{1}{N}\sum_{k=1}^{N}\left|x-\hat{x}\right| \end{array}$$
with *x* the reference, $\hat{x}$ the estimate *N*, the number of samples.*SCORE:* Number of simulations in the test dataset where the considered method obtains the smallest RMSE.

The average of each criteria are evaluated on the dataset as:(14)$$\begin{array}{c} C_{\chi }=\frac{1}{N_{sim}}\sum_{k=1}^{N_{sim}}\chi_{k} \end{array}$$
with χ the selected evaluation criterion and $N_{sim}$ the number of simulations in the test dataset.

$C_{MAE}$ Criterion analysis: Figure 11 presents the MAE average $C_{MAE}$, evaluated on the whole test dataset for $LSTM_{IMU,\;V_{1}}$, $LSTM_{IMU,\;V_{6}}$, $LSTM_{IMU\;DYN,\;V_{1}}$, $LSTM_{IMU\;DYN,\;V_{6}}$ and the Dead-Reckoning.

The LSTMs accurately estimate position and velocity, both with *IMU* and *IMU DYN measurement*, with errors around a few meters for the positions. This criterion confirms that the local navigation frame rotation improves $p_{z}$, $v_{x}$ and $v_{z}$ estimation for LSTMs trained with *IMU measurement*, and $p_{y}$, $v_{x}$ and $v_{y}$ estimation for LSTMs trained with *IMU DYN measurement*. As expected, LSTMs are not adapted to estimate the projectile roll angle contrary to the pitch and the yaw angle.

$C_{SCORE}$ Criterion analysis: Figure 12 presents the score, $C_{SCORE}$, evaluated on the whole test dataset for $LSTM_{IMU,\;V_{1}}$, $LSTM_{IMU,\;V_{6}}$, $LSTM_{IMU\;DYN,\;V_{1}}$, $LSTM_{IMU\;DYN,\;V_{6}}$ and the Dead-Reckoning.

According to Figure 12, an LSTM is an accurate approach to estimate projectile position and velocity in a GNSS-denied environment. However, a LSTM is not the optimal method for the orientation estimation. Furthermore, an LSTM is able to generalize the learned features over a large projectile fire dataset as well as learn and predict trajectories from different sensor models. Finally, $C_{MAE}$ and $C_{SCORE}$ analysis confirms that the local navigation frame rotation is an appropriate method to optimize $p_{z}$, $v_{x}$ and $v_{z}$ for LSTMs trained with *IMU measurement*, and $p_{y}$, $v_{x}$, and $v_{y}$ for LSTMs trained with *IMU DYN measurement*.

#### 5.2.4. Errors at Impact Point

This section focuses on the errors at the impact point, i.e., position errors $\left(p_{x},p_{y}\right)$ at the final time of a shot. Figure 13a shows the impact point errors of LSTMs with *IMU* and *IMU DYN measurements* and the Dead-Reckoning. Figure 13b presents where the impact point errors are located in different error zones.

Whatever the inertial sensor model, The Dead-Reckoning impact point errors are greater than 100 m, contrary to LSTMs. Focusing on *IMU measurements*, the majority of $LSTM_{IMU,\;V_{6}}$ (rotation) impact point errors are less than 5 m contrary to $LSTM_{IMU,\;V_{1}}$ (no rotation) where errors are lower than 20 m. Thus, the local navigation frame rotation allows to strongly reduce errors at the impact point. Focusing on *IMU DYN measurements*, the local navigation frame rotation deteriorates estimation accuracy. Indeed, 266 simulations have impact point errors less than 5 m with $LSTM_{IMU\;DYN,\;V_{1}}$ while 180 simulations have impact point errors less than 5 m with $LSTM_{IMU\;DYN,\;V_{6}}$.

In summary, an LSTM is an accurate approach to estimate projectile position as errors are less than ten meters in a GNSS-denied environment. These results are comparable to those obtained by commercial GNSS-guided mortars. Moreover, the local navigation frame rotation is useful with *IMU measurement* and allows to minimize position errors.

## 6. Conclusions

This paper presents a deep learning approach to estimate a gun-fired projectile trajectory in a GNSS-denied environment. Long-Short-Term-Memories (LSTMs) are trained from the embedded IMU, the magnetic field reference, flight parameters specific to the projectile (the fin angle, the initial velocity, the barrel elevation angle) and a time vector. The impact of three preprocessing methods are analyzed: the input data normalizations, the local navigation frame rotation and the inertial sensor model. According to the reported results, the LSTMs accurately estimate projectile positions and velocities compared to a conventional navigation algorithm as errors are around ten meters, similar to the GNSS-guided projectile accuracy. Nevertheless, LSTM suffers in the estimation of the projectile orientation, especially for the high dynamic roll angle. The input data normalization provides no interesting results while the local navigation frame rotation optimizes the position and velocity estimation. Moreover, the results prove that the LSTMs generalize the learned features on large datasets independently of the inertial sensor model considered. Based on results reported in this paper, the next step is to implement a deep Kalman filter by considering the LSTM predictions as observations.

## Figures and Tables

**Figure 1 sensors-23-03025-f001:**
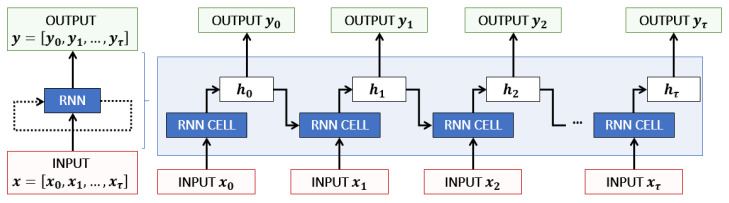
RNN layer overview, roll and unroll: many-to-many representation.

**Figure 2 sensors-23-03025-f002:**
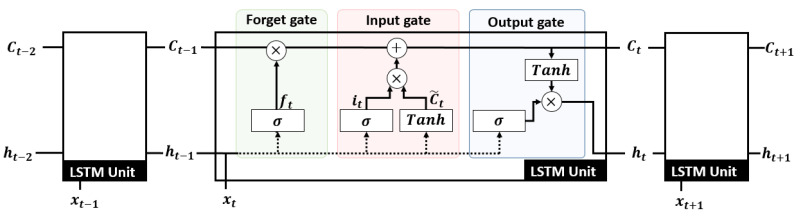
LSTM cell operating principle composed by three gates.

**Figure 3 sensors-23-03025-f003:**
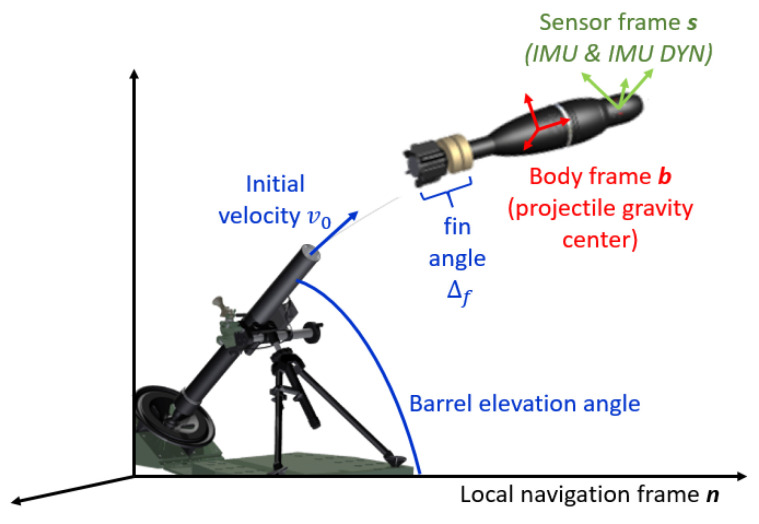
Navigation frames (black—local navigation frame ***n***, red—body frame ***b***, green—sensor frame ***s***) and flight parameters for a finned projectile (fin angle $\delta_{f}$, initial velocity $v_{0}$, barrel elevation angle α).

**Figure 4 sensors-23-03025-f004:**
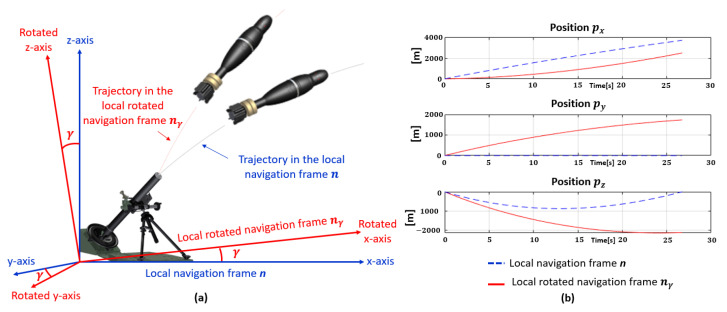
(**a**) Local navigation frame n and local rotated navigation frame $n_{\gamma }$. (**b**) Projectile position in the local navigation frame ***n*** (blue dashed line), projectile position in the local rotated navigation frame $n_{\gamma }$ (red solid line).

**Figure 5 sensors-23-03025-f005:**
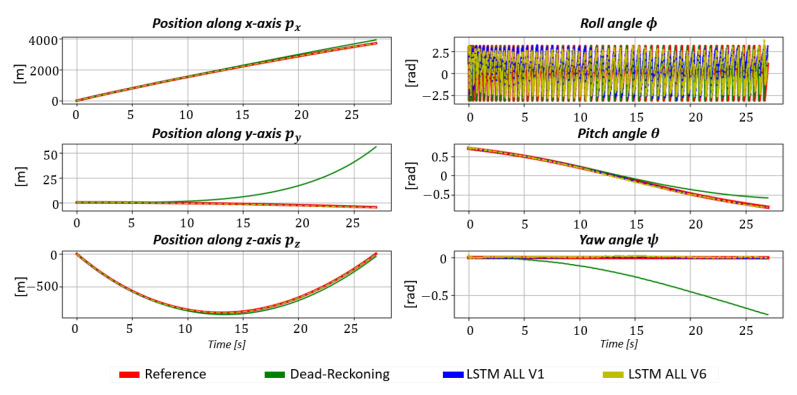
Estimated projectile position [m] and Euler angles [rad] obtained by the Dead-Reckoning (green), $LSTM_{ALL,\;V_{1}}$ (blue), $LSTM_{ALL,\;V_{6}}$ (yellow) and the reference (red).

**Figure 6 sensors-23-03025-f006:**
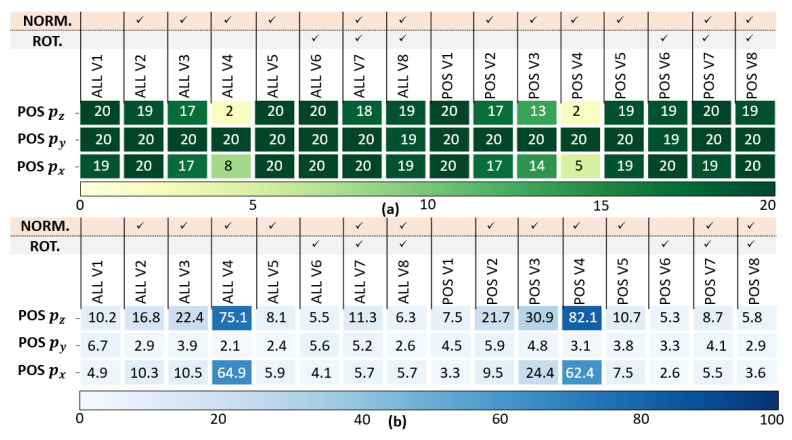
Position analysis: (**a**) Success Rate $C_{1}$, (**b**) Error Rate $C_{2}$ (in %) of $LSTM_{ALL,\;V_{1}-V_{8}}$ and $LSTM_{POS,\;V_{1}-V_{8}}$.

**Figure 7 sensors-23-03025-f007:**
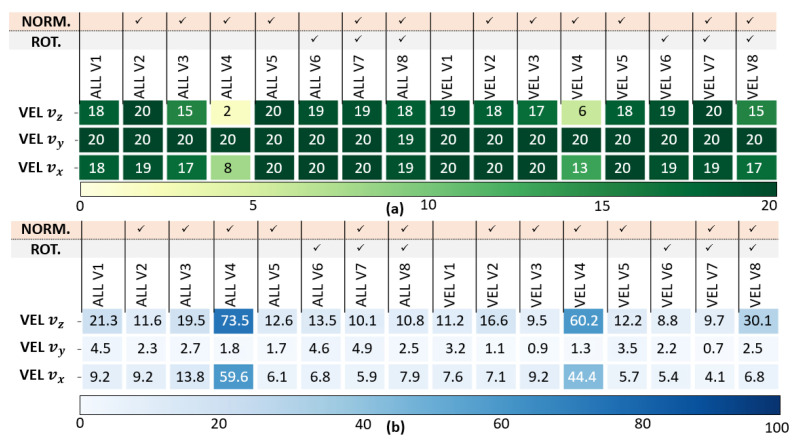
Velocity analysis: (**a**) Success Rate $C_{1}$, (**b**) Error Rate $C_{2}$ (in %) of $LSTM_{ALL,\;V_{1}-V_{8}}$ and $LSTM_{VEL,\;V_{1}-V_{8}}$.

**Figure 8 sensors-23-03025-f008:**
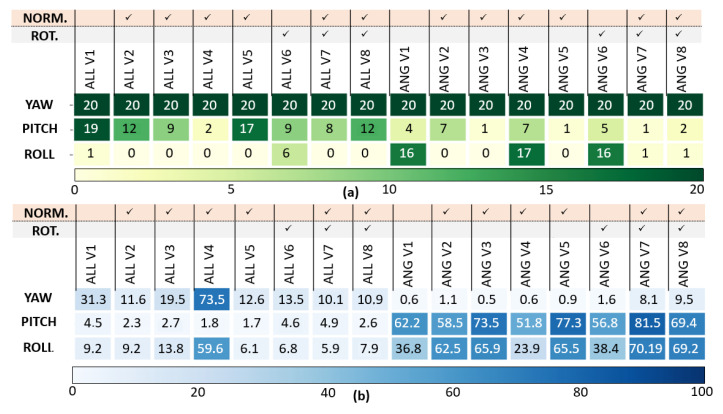
Orientation analysis: (**a**) Success Rate $C_{1}$, (**b**) Error Rate $C_{2}$ (in %) of $LSTM_{ALL,\;V_{1}-V_{8}}$ and $LSTM_{ANG,\;V_{1}-V_{8}}$.

**Figure 9 sensors-23-03025-f009:**
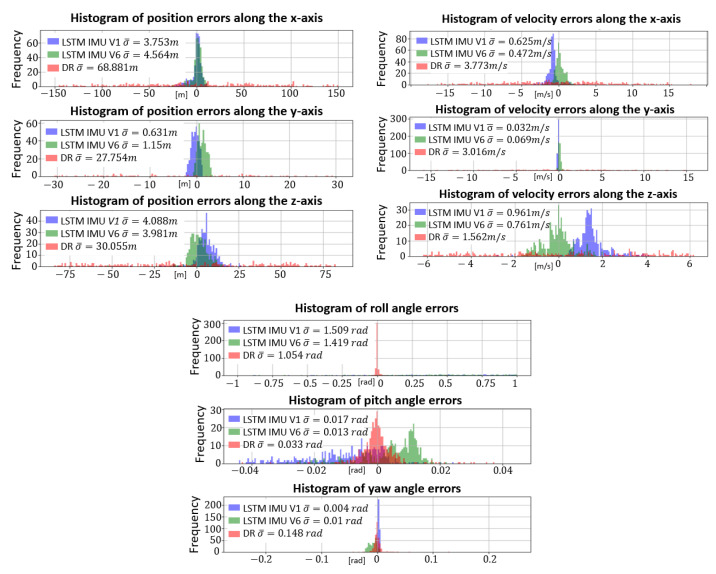
Average position, velocity and orientation error histogram obtained by $LSTM_{IMU,\;V_{1}}$ (blue), $LSTM_{IMU,\;V_{6}}$ (green) and Dead-Reckoning (red).

**Figure 10 sensors-23-03025-f010:**
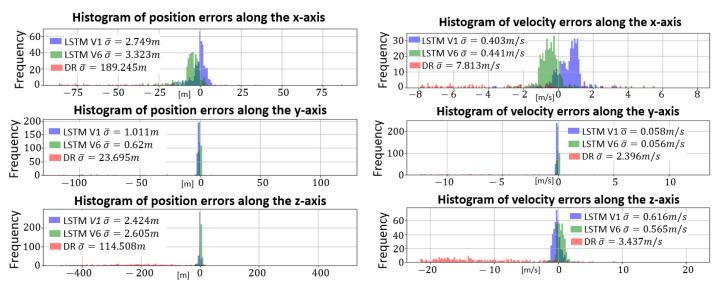
Average position, velocity and orientation error histogram obtained by $LSTM_{IMU\;DYN,\;V_{1}}$ (blue), $LSTM_{IMU\;DYN,\;V_{6}}$ (green) and Dead-Reckoning (red).

**Figure 11 sensors-23-03025-f011:**
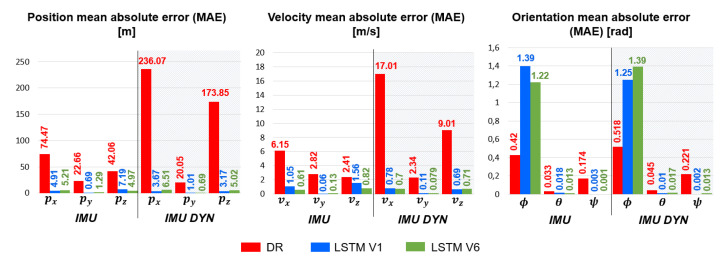
Position, velocity and orientation MAE average $C_{MAE}$ obtained by $LSTM_{IMU,\;V_{1}}$, $LSTM_{IMU,\;V_{6}}$, $LSTM_{IMU\;DYN,\;V_{1}}$, $LSTM_{IMU\;DYN,\;V_{6}}$ and the Dead-Reckoning.

**Figure 12 sensors-23-03025-f012:**
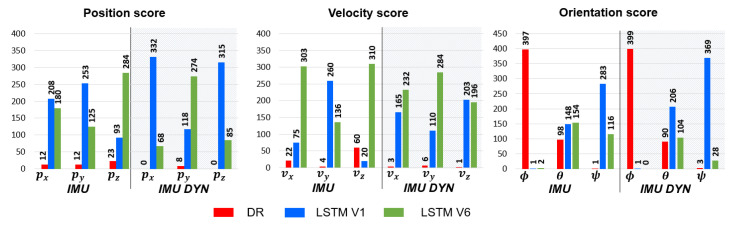
Position, velocity and orientation score obtained by $LSTM_{IMU,\;V_{1}}$, $LSTM_{IMU,\;V_{6}}$, $LSTM_{IMU\;DYN,\;V_{1}}$, $LSTM_{IMU\;DYN,\;V_{6}}$ and the Dead-Reckoning.

**Figure 13 sensors-23-03025-f013:**
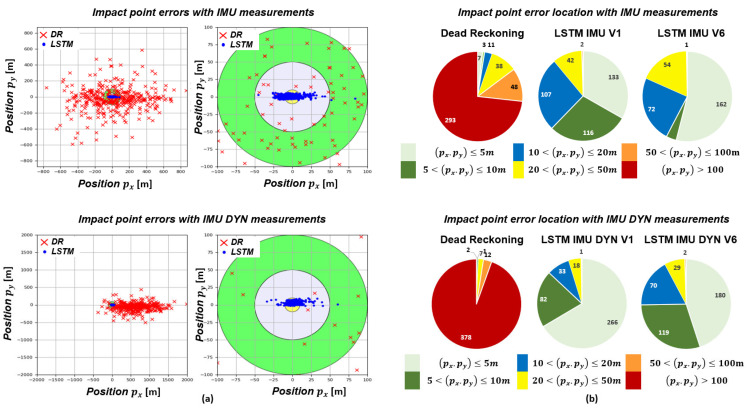
(**a**) Errors at impact point obtained by $LSTM_{IMU,\;V_{1}}$ and $LSTM_{IMU\;DYN,\;V_{1}}$ (blue cross), $LSTM_{IMU,\;V_{6}}$ and $LSTM_{IMU\;DYN,\;V_{6}}$ (green cross) and the Dead-Reckoning (red dot). (**b**) Impact point error location.

**Table 1 sensors-23-03025-t001:** Version specifications: Influence of the normalization and the local navigation frame rotation.

Name	Normalization	Rotation
$V_{1}$	No	No
$V_{2}$	MM(T),MM(M),MM(P)	No
$V_{3}$	MM(T,M,P)	No
$V_{4}$	STD(T),STD(M),STD(P)	No
$V_{5}$	STD(T,M,P)	No
$V_{6}$	No	Yes
$V_{7}$	MM(T,M,P)	Yes
$V_{8}$	STD(T,M,P)	Yes

**Table 2 sensors-23-03025-t002:** Training characteristics of $LSTM_{ALL,\;POS,\;VEL,\;ANG,\;V_{1-8}}$.

*Dataset*	Training Dataset:	100 Simulations (Validation: 10 Simulations)
	Test Dataset:	20 Simulations
*Input data*	Batch size:	64 ($S_{eq\;len}$: 20 timesteps)
	Input data:	$I_{n\;Features}=\left(M,P,T\right)\in R^{16}$ (with *IMU measurements*)
	Cost function:	Mean Squared Error (MSE)
*Training*	Optimization algorithm:	ADAM [41] (Learning rate : $1\times 10^{-4}$)
	LSTM layer:	2 (Hidden units: 64–128)

**Table 3 sensors-23-03025-t003:** Training characteristics of $LSTM_{IMU,\;V_{1}}$, $LSTM_{IMU\;DYN,\;V_{1}}$, $LSTM_{IMU,\;V_{6}}$, $LSTM_{IMU\;DYN,\;V_{6}}$.

*Dataset *	Training Dataset:	4000 Simulations (Validation: 400 Simulations)
	Test Dataset:	400 Simulations
*LSTM name*	No normalization	$LSTM_{IMU,\;V_{1}}$ (with *IMU measurements*)
	& No rotation	$LSTM_{IMU\;DYN,\;V_{1}}$ (with *IMU DYN measurements*)
	No normalization	$LSTM_{IMU,\;V_{6}}$ (with *IMU measurements*)
	& Rotation	$LSTM_{IMU\;DYN,\;V_{6}}$ (with *IMU DYN measurements*)
*Input data*	Batch size:	64 ($S_{eq\;len}$: 20 timestamp)
	Input data:	$I_{n\;Features}=\left(M,P,T\right)\in R^{16}$
	Cost function:	Mean Squared Error (MSE)
*Training*	Optimization algorithm:	ADAM [41] (Learning rate: 1 × $10^{-4}$)
	LSTM layer:	2 (Hidden units: 64–128)

## Data Availability

Not applicable.

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
