# Peer review of "LSTM-Based Projectile Trajectory Estimation in a GNSS-Denied Environmentâ€"

_sensors, 2023, doi:10.3390/s23063025_

Round 1
Reviewer 1 Report
The authors propose an LSTM-based deep learning method to estimate projectile trajectories in a GNSS-denied environment. The authors focus on LSTM input data preprocessing (including normalization and navigation frame rotation) and the effect of sensor error models on the accuracy of projectile trajectory prediction. The algorithm is compared with a classical Dead-Reckoning algorithm through various aspects such as different error criteria and the position error of the impact point. The results show that the present method is superior in the estimation of projectile position and velocity with less error compared to the classical method.
This paper is scientific and rational, with sufficient experiments to verify the accuracy of the method through qualitative and quantitative experiments. It is suggested that the improvement over the traditional method should be clearly reflected by specific data in the abstract section.
2. In order to help the authors improve their manuscript, and inform the
Academic Editor's decision, please consider providing some additional,
specific comments such as:
1. What is the main question addressed by the research?
The authors propose an LSTM-based deep learning method to estimate projectile trajectories in a GNSS-denied environment.
2. Do you consider the topic original or relevant in the field? Does it address a specific gap in the field?
Yes,I think the topic is relevant to the field. This paper provides an idea or method for this field.
3. What does it add to the subject area compared with other published material?
The authors focus on LSTM input data preprocessing (including normalization and navigation frame rotation) and the effect of sensor error models on the accuracy of projectile trajectory prediction.
4. What specific improvements should the authors consider regarding the methodology? What further controls should be considered?
It is suggested that the improvement over the traditional method should be clearly reflected by specific data.
5. Are the conclusions consistent with the evidence and arguments
presented and do they address the main question posed?
Yes.
6. Are the references appropriate?
Yes.
7. Please include any additional comments on the tables and figures.
The tables and figures are more informative and convincing.
Author Response
Point 1: This paper is scientific and rational, with sufficient experiments to verify the accuracy of the method through qualitative and quantitative experiments. It is suggested that the improvement over the traditional method should be clearly reflected by specific data in the abstract section.
Response 1: Thank you for your review and feedback.
The improvement of the proposed solution compared to the traditional method has been added in the abstract part (see lines 11-12).
Reviewer 2 Report
Great paper. The problem is well-defined, the approach is robust, and the discussion of the results is comprehensive. My only complaint is that some of the figures are difficult to read because of their size - specifically, the top right of Fig 5 and the numbers in Figures 6-8.
Author Response
Point 1: Great paper. The problem is well-defined, the approach is robust, and the discussion of the results is comprehensive. My only complaint is that some of the figures are difficult to read because of their size - specifically, the top right of Fig 5 and the numbers in Figures 6-8.
Response 1: Thank you for your review and feedback.
Figures 5, 6, 7 and 8 have been enlarged and modified to be more readable.
Reviewer 3 Report
This paper highlights the utilization of the deep learning approach to estimate a projectile trajectory in a GNSS-denied environment. There are six contributions listed by the authors, ranging from detailing the LTSM-based approach to estimate the projectile positions, velocities and Euler angles, up to comparing the LSTM accuracy to a dead-reckoning.
The paper has been well-prepared in a well-structured manner. The presented literature review is good with an intensive review of all related references. The cited references used are also the most recent. In the list, 21 out of 40 cited references were published within the last five years. Some minor modifications are suggested for Section 2.2 (lines 95 to 112). Refer text for the suggestions.
The methodology of the work in terms of the problem formulation and LSTM input data prepossessing are well presented in Sections 3 and 4, respectively. They are clear, repeatable and scientifically sound. The depicted tables and figures of the results presented in Section 5 are all vivid and easy to read. The discussion of the results was also thorough and sufficient.
I would recommend that this paper be accepted in its present form, with minor improvement in section 2.2.

Author Response
Point 1: "I would recommend that this paper be accepted in its present form, with minor improvement in section 2.2."
Response 1: Thank you for your review and feedback.
References for each use of AI in the military field have been added in section 2.2 (see lines 95-112).
Reviewer 4 Report
This paper presents a deep-learning approach to estimate a gun-fired projectile trajectory in a GNSS-denied environment.
1) In line 202, what's the meaning of c? Is it a typo?
2) In line 265, V_7, V_8, I think it may be a typo.
3) In line 266, how to do normalization for all features? Please give the specific equations.
4) Table 2 shows that the seq_len of input data is 20 timestamps. Why is the time longer than 25s?
5) In section 5, please add the results with different seq_len of input data.
Author Response
Thank you for your review and feedback.
Point 1: In line 202, what's the meaning of c? Is it a typo?
Response 1: This is a typo that has been corrected in the document (see line 202).
Point 2: In line 265, V_7, V_8, I think it may be a typo.
Response 2: This is a typo that has been corrected in the document (see line 265).
Point 3: In line 266, how to do normalization for all features? Please give the specific equations.
Response 3: Equations allowing to determine the normalization factor for all features have been added in the document (see lines 265-271).
Point 4: Table 2 shows that the seq_len of input data is 20 timestamps. Why is the time longer than 25s?
Response 4: The sequence length seq_len is set to 20 time steps, which corresponds to 20 samples representing a window of 0,02s as the sensor sampling period is 0.001s. An explication about the size of seq_len has been added to the document (see lines 314-315).
Point 5: In section 5, please add the results with different seq_len of input data.
Response 5: The parameter seq_len is adjusted according to the input data used. This parameter has been determined empirically in order to obtain the best estimation performances.
Indeed, if the sequence is too short, the LSTM performances are degraded as it becomes dependent on the sensor noises (accelerometer, gyrometer, and magnetometer).
If the sequence is too long, the performance is barely improved except that the training becomes too expensive. Moreover, for a real time implementation, a too long sequence requires very long time responses.
Due to these reasons, results are presented for seq_len=20 which corresponds to the best results obtained.